# Advances in the Management of Lung Cancer Brain Metastases

**DOI:** 10.3390/cancers16223780

**Published:** 2024-11-09

**Authors:** Kathryn G. Hockemeyer, Chad G. Rusthoven, Luke R. G. Pike

**Affiliations:** 1Department of Radiation Oncology, Memorial Sloan Kettering Cancer Center, New York, NY 10065, USA; 2Department of Radiation Oncology, University of Colorado, Aurora, CO 80045, USA

**Keywords:** brain metastases, lung cancer, stereotactic radiosurgery

## Abstract

Brain metastases are common in patients with lung cancer, leading to significant neurological injury and neurologic mortality. Recent research efforts in radiation and systemic therapies have led to significant improvements in clinical outcomes for these patients and have shed light on possible novel combinatorial approaches. Many patients will develop brain metastases while on systemic therapy and it is critical to define how to integrate radiation therapy, the current standard of care, with treatments that have efficacy against brain metastases. Herein, we review the historical context and recent advances in radiation therapy and systemic therapies. We highlight ongoing approaches to integrate multimodal regimens and discuss novel strategies to predict brain metastases to further guide management.

## 1. Introduction

The majority of lung cancers are non-small cell lung cancer (NSCLC), of which lung adenocarcinoma and squamous cell carcinoma are the most common histological subtypes [1]. NSCLC, especially lung adenocarcinoma, is characterized by a high frequency of driver mutations in oncogenes including *EGFR*, *KRAS*, *STK11*, *MET*, *ALK*, *ERBB2/3*, *FGFR1/2/3*, *ROS*, and *RET* [1]. Patients with NSCLC with oncogene driver mutations have rates of brain metastases (BrMs) in the range of 30–40% and up to 60% of these patients will develop BrMs [2,3,4,5]. Small cell lung cancer (SCLC), which makes up approximately 15% of lung cancer diagnoses, is characterized by a propensity for early metastasis with only one-third of patients presenting with early-stage disease [6]. SCLC tumors are characterized by frequent inactivating mutations in *TP53* and *RB1* and significant intratumoral heterogeneity, which contributes to metastasis and resistance [6]. SCLC is associated with a high propensity to develop BrMs; 10–25% of patients present with BrMs, and up to 60% of patients will develop BrMs [7,8]. Substantial advances have been made in the management of BrMs for lung cancer. We review the advances in radiation therapy, which continues to be a mainstay in the management of BrMs in lung cancer patients. Clinical trials have expanded the application of stereotactic radiosurgery (SRS), and evaluated strategies for preserving neurocognition with whole-brain radiotherapy (WBRT), including in prophylactic cranial irradiation (PCI) [9]. In addition, we also discuss emerging systemic therapies that have been found to have intracranial efficacy, albeit predominantly in post hoc and retrospective analyses.

## 2. Radiotherapy

WBRT [10] and SRS [11] were first described in the 1950s for targeting lesions in the brain. Given that the majority of BrMs result from the hematogenous spread of tumor cells, it was thought that the brain could be seeded with micrometastases at the diagnosis of BrMs [12,13]. Thus, WBRT was preferred for the management of single or multiple BrMs to prevent neurologic compromise from uncontrolled brain tumors [13]. Indeed, WBRT was shown to provide clinical improvement [10,14] as well as a possible survival benefit in select patients [15], leading to its adoption as standard practice for the management of brain metastases in NSCLC. Randomized prospective trials subsequently evaluated the addition of surgery [16,17] or SRS [18,19] to WBRT in patients with three or fewer BrMs and found improved survival and local control with the addition of either modality to WBRT relative to WBRT alone. Given the neurocognitive toxicity associated with WBRT, subsequent phase III clinical trials challenged WBRT as the standard of care for patients with 1–3 or 4 BrMs by comparing SRS with or without WBRT [20,21,22,23]. These trials have demonstrated improved central nervous system (CNS) disease control with the addition of WBRT, but improved quality of life and cognitive preservation with SRS alone, and no difference in overall survival, thus establishing SRS as the standard of care for limited BrMs, often defined historically as up to four lesions based on the inclusion criteria of these trials [20,21,22,23,24]. In a prospective study of 1194 patients, both the OS and rate of adverse events were comparable between patients with 2–4 lesions and those with 5–10 [25]. These results expanded the use of SRS for higher numbers of BrMs and decreased the reliance on a strict cutoff of four or less BrMs in guidelines and clinical practice. A subsequent metanalysis of 54 trials of 11,898 patients demonstrated that the combination of SRS and WBRT was superior to WBRT alone for local control while WBRT and SRS exhibited improved distant brain control over SRS alone [26]. No differences were found in the overall survival and the addition of WBRT was associated with worse neurocognitive outcomes [26]. It is important to note that the clinical trials evaluating radiotherapy in BrMs comprised 40–100% NSCLC patients, but also included additional cancer types, including breast, prostate, kidney, and colorectal cancer. Nonetheless, these trials have guided clinical practice recommendations in NSCLC [27]. In NSCLC, current emerging advancements in WBRT aim to preserve cognitive function whereas efforts to improve SRS are evaluating an increasing number of target lesions, fractionation, neoadjuvant vs. adjuvant approaches when combined with surgery, and optimal integration with emerging systemic therapies including immunotherapy and target therapies.

SCLC has high rate of metastasis to the brain, with 40–60% of patients developing BrMs during the course of their disease [28,29,30]. Given the propensity of SCLC to develop metastases and the poor prognosis of SCLC BrMs, prophylactic cranial irradiation (PCI) (which functionally entails the prophylactic administration of WBRT for the goal of eliminating potential subclinical, microscopic disease in the brain) has been the standard since studies established a local control and survival benefit in the 1990s and 2000s [31,32,33]. Due to the high propensity of SCLC to develop BrMs, concerns for diffuse, multicentric brain lesions, management has favored WBRT for the treatment of possible micrometastases. For SCLC patients with BrMs, WBRT thus continues to be a mainstay in the treatment of SCLC patients, who were excluded from the landmark prospective trials establishing SRS for limited BrMs. However, restrospective studies have found similar survival rates between SRS and WBRT; ongoing phase II and III trials are investigating if SRS is comparable to WBRT in SCLC patients with BrMs. For PCI, randomized studies of SCLC are currently elucidating if active surveillance and PCI are comparable and if hippocampal avoidance (HA) can reduce the cognitive decline from PCI. Table 1 includes relevant clinical trials leading to advances in radiotherapy for the management of BrMs and the current standard of care.

### 2.1. Whole-Brain Radiotherapy

WBRT was a mainstay for the treatment of NSCLC BrMs after it was described to provide symptomatic improvement in the 1950s and 1960s [10,14]. Prior to the era of routine imaging for monitoring tumor growth, dose escalation trials established regimens of 20–30 gray (Gy) in 5–10 fractions for achieving clinical remission in patients with BrMs, including those with lung cancer [34,35]. However, the role of WBRT has been decreasing in more recent years due to concerns regarding toxicity, the lack of a clear OS benefit with WBRT for most patients, and the emergence of alternative treatment options for patients with increasing numbers of BrMs [36]. Indeed, the Alliance N0574 trial, which showed a difference in the overall survival between SRS with WBRT vs. SRS alone, found that the addition of WBRT to SRS led to worsened cognitive deterioration at 3 months compared to that with SRS alone (91.7% vs. 63.5%, *p* < 0.001) and a decreased QoL score (−10.9 vs. −1.3 mean change from baseline, *p* = 0.002) [23]. The QUARTZ trial randomized 538 patients with NSCLC who were determined by a multidisciplinary team of neurosurgeons and radiation oncologists to be poor candidates for surgery or SRS to WBRT or best supportive care, including dexamethasone [37]. The trial did not demonstrate significant improvements with WBRT in the overall survival, quality of life, or quality-adjusted life-years. The overall survival was comparable between the two groups, with a median survival of approximately 9 weeks. The subgroup analysis of the QUARTZ trial demonstrated a survival benefit for patients less than 60 years of age and a trend towards improved survival in patients with a Karnofsky performance score (KPS) of 70 or greater and controlled extracranial disease [37], suggesting that certain populations may benefit from WBRT.

In NSCLC patients who have a good performance status, but present with BrMs that are too extensive for targeting with SRS, WBRT remains a reasonable option for local control and palliation. Various strategies have been developed for improving cognitive preservation with WBRT [9]. The phase III RTOG 0614 trial evaluated the potential neurocognitive protective effect of memantine, an NMDA receptor antagonist, in the context of WBRT [38]. Memantine was associated with a significantly longer time to cognitive decline (HR 0.78; 95% CI 0.62–0.99, *p* = 0.01) and a trend towards less decline in the primary endpoint of delayed recall (0 vs. −0.90, *p* = 0.059) relative to WBRT alone [38]. To determine if HA using intensity-modulated radiotherapy (IMRT) could preserve cognition, the phase III NRG CC001 trial compared HA-WBRT to WBRT, with both arms receiving memantine [39]. The time to cognitive failure was significantly reduced in HA-WBRT with memantine relative to WBRT with memantine (adjusted HR 0.75, 95% CI 0.58–0.95, *p* = 0.02), establishing HA-WBRT with memantine as the standard of care for patients with no metastases in the hippocampal region. In addition, The National Taiwan University Hospital conducted a phase II randomized trial to investigate if hippocampal avoidance (HA) could preserve neurocognition in WBRT [40]. With no difference in the PFS or OS, HA yielded an improved recognition–discrimination index and memory although there were no significant differences in the total recall or delayed recall.

In SCLC, the recognition of high rates of CNS progression led to the development of PCI as a strategy to mitigate BrMs. Although PCI was effective in reducing the rates of BrMs, concerns emerged regarding the potential toxicity to cognitive function and QOL [41]. PCI became the standard of care for LS-SCLC following a 1999 meta-analysis of PCI in patients predominantly with LS-SCLC in remission, demonstrating a reduced incidence of BrMs from 58.6% to 33.3% with PCI, associated with an OS benefit of 5.4% at 3 years [32]. In 2007, a phase III EORTC trial randomized patients with ES-SCLC to PCI or no further therapy and identified a 14% benefit in the OS at 1 year after randomization [33]. As a result, PCI became the standard of care for both LS and ES-SCLC. However, a pooled analysis of the PCI RTOG 0212 and 0214 trials has shown cognitive decline with PCI [42].

In an effort to reduce the deleterious cognitive effects of PCI, the PREMER trial randomized patients with SCLC to PCI or PCI with hippocampal avoidance (HA-PCI) [43]. HA-PCI was protective against cognitive decline compared to standard PCI with no differences in the OS or quality of life. A separate trial sponsored by the Netherlands Cancer Institute (NKI) randomized SCLC patients to PCI or HA-PCI and found the decline in the Hopkins verbal learning test score to be comparable between the two groups [44]. The NRG CC003 (NCT02635009) phase III trial evaluating intracranial response and rates of neurocognitive failure with PCI vs. HA-PCI was recently reported [45]. The addition of HA to PCI led to reduced neurocognitive failure (adjusted HR 0.77, 95% CI 0.61–0.98, *p* = 0.03) with a non-inferior 12-month intracranial response rate [45].

Other trials have sought to improve QOL outcomes in SCLC by re-examining the role of PCI altogether in the era of brain MRI surveillance. In a Japanese randomized trial, patients with ES-SCLC without BrMs on MRI were randomized to prophylactic cranial radiation at 25 Gy in 10 fractions or observation, with all patients undergoing routine MR brain imaging, and observation was found to yield a non-inferior OS compared to that with PCI (13.7 months vs. 11.6 months, *p* = 0.094) [46]. The ongoing phase III Maverick (NCT04155034) and EORTC PRIMALung trials [47] are randomizing patients with LS-SCLC and ES-SCLC to MR surveillance with or without PCI.

### 2.2. Stereotactic Radiosurgery

After it was demonstrated that SRS yielded reduced neurocognitive toxicity with no difference in the overall survival for patients with up to 4 BrMs [20,21,22,23] and subsequently in patients with 5–10 BrMs [25], research efforts have sought to advance SRS technology for multicentric BrMs and elucidate optimal fractionation and combination with surgery and systemic therapies. With the advent of volumetric modulated arc therapy, it became feasible to target multiple lesions using a single isocenter, which can reduce the treatment time of SRS for BrMs [48,49]. A retrospective study of 173 patients with 1014 BrMs showed a median beam-on time of 4.1 min and a median brain dose of 2.19 Gy; the local control at 1 year was 99%, while grade 2 or higher radionecrosis at 1 year was 1.4% [50]. These findings suggest that single-isocenter multicentric SRS technology is both effective and safe, expanding the feasibility of targeting higher numbers of BrMs in a reasonable amount of time for patients.

To mitigate the potential late side effect of radiation necrosis, a number of studies have evaluated the potential benefits of fractionating SRS over multiple treatments. Retrospective and dose-escalation studies suggest 27 Gy in three fractions to be associated with improved local control, with rates of radionecrosis at around 10% (95% CI 0–29%) [51,52]. However, for small lesions (≤2 cm), 15–24 Gy in one fraction is a reasonable approach based on the RTOG 90-05 trial, with doses varying by patient-specific and tumor-specific factors [27,53]. A meta-analysis of 24 studies of single-fraction vs. multifraction SRS revealed multiple fractions were associated with decreased rates of radionecrosis for tumor size 4–14 cc and 2–3 cm diameter [54]. An NRG phase III trial is examining one or three fractions of SRS in patients with one to eight lesions with a maximum diameter between 1.0 cm and 3.0 cm (NCT06500455).

An active area of research with respect to SRS is characterizing the safety and efficacy of SRS for an increasing number of lesions. A phase III trial, which was closed early due to slow accrual, randomized 72 patients with 4–15 BrMs to SRS or WBRT and demonstrated that SRS preserved cognitive function with no differences in the local control or OS between the two groups [55]. A retrospective analysis of patients with *EGFR*-mutated and *ALK*-rearranged NSCLC receiving SRS for four or more BrMs found that the mean hippocampal (1.2 Gy) and whole-brain (0.8 Gy) doses remained low with SRS for more than ten lesions [56]. Another retrospective study of patients treated with SRS for 10 or more lesions without concurrent WBRT found freedom from progression at 12 months of 96.8% for upfront treatment and 83.6% for salvage treatment with a mean hippocampal dose of 1.5 Gy [57]. In addition, a prospective cohort of 70 patients with 1174 BrMs in which 73% of the patients received a single fraction of 20–24 Gy found a median OS of 19.2 months, with local control in 97.3% of the lesions and a cumulative incidence of radionecrosis in 2.1% of the lesions [58]. These findings together suggest that SRS for many BrMs may achieve higher hippocampal sparing than in HA-WBRT. Ongoing randomized clinical trials are currently evaluating SRS for up to 20–30 BrMs, comparing SRS to HA-WBRT (NCT03550391, NCT03075072, NCT04277403) or comparing SRS with or without WBRT (NCT03775330).

In an effort to improve CNS disease control without WBRT, the phase 3 METIS trial examined tumor treating fields (electric fields that disrupt cancer cell division) in addition to SRS. In patients with mutation-negative NSCLC, tumor treating fields (TTFs) led to a doubling of the intracranial PFS (median of 21.9 vs. 11.3 months, *p* = 0.02) without impairing cognition [59]. These findings suggest that TTFs could be an alternative to WBRT with less cognitive toxicity in patients with extensive BrMs, although this has not been explored in the clinical trial setting.

In SCLC, due to the high rate of CNS metastasis in SCLC and concerns for diffuse or multifocal BrMs, as well as the prevalence of PCI as a component of first-line therapy, patients with BrMs were excluded from the randomized trials that established SRS as the first line for BrMs [60]. Consequently, WBRT remained the standard of care, even in the context of limited or solitary BrMs [60]. To evaluate the potential role of SRS alone in SCLC, the FIRE-SCLC cohort study compared 710 SCLC patients treated with first-line SRS between 1994 and 2008 to a cohort of patients receiving first-line WBRT [31]. Although WBRT was associated with an improved time to CNS progression (HR 0.38, 95% CI 0.26–0.55, *p* < 0.001), this did not translate into an improvement in the OS with WBRT (6.5 months for SRS, 5.2 months for WBRT, *p* = 0.003) [31]. Subsequently, a systematic review and meta-analysis of retrospective studies comparing SRS and WBRT in SCLC patients with BrMs found a similar OS between the two groups [61]. Together, these data suggest that SRS may be an appropriate first-line therapy in some patients [60]. The phase II ENCEPHALON (NCT03297788) randomized 56 patients with SCLC, up to 10 BrMs, and a KPS greater than 50 to WBRT or stereotactic radiotherapy (SRT, SRS for small lesions 20 G or 18 Gy, and hypofractionated SRT to 30 Gy in five fractions for lesions > 3 cm). The primary endpoint of neurocognition after cerebral irradiation was 7.7% in the SRT group and 24% in the WBRT group (*p* = 0.0723), whereas there were no differences in the overall survival between the two groups [62]. An additional phase II prospective single-arm trial of 100 patients across four centers evaluated SRS/SRT in SCLC patients with up to 10 BrMs; 66% of the patients were treated with SRS alone and 32% underwent SRT for at least one site. The rate of neurologic death at 1 year was 11.0% (95% CI 4.8–17.2%), which is comparable to the authors’ historical institutional rate of 17.5% for neurologic death with WBRT at 1 year [63]. The phase III trials NRG CC009 (NCT04804644) and NCT06457906 comparing WBRT to SRS for patients with SCLC and up to 10 BrMs are ongoing and will likely provide a definitive answer if SRS and/or SRT is a safe therapeutic approach for SCLC patients with BrMs.

### 2.3. Surgery and Radiotherapy

Surgery remains an important component in the local management of large and/or symptomatic BrMs for the management of mass effect or symptoms and for diagnostic information [27]. Recurrence at the resection bed 1 year after surgery alone is relatively common, with rates of approximately 50–60% in clinical trials [22,64]. Adjuvant WBRT and SRS have been demonstrated to significantly improve local control at the resected site [17,22,64], although, as expected, WBRT but not SRS has been shown to improve intracranial recurrence [64,65]. The EORTC 22952-26001 trial demonstrated that after SRS or surgery, adjuvant WBRT improved the local control and time to neurologic death without improving the performance status [22]. A phase III trial of 194 patients comparing post-operative WBRT to SRS showed decreased cognitive deterioration with SRS relative to WBRT (52% vs. 85% at 6 months, *p* < 0.00031), while WBRT was associated with improved intracranial disease control (55% vs. 81% at 6 months) [65,66]. In addition, a retrospective cohort study of 558 patients with resected BrMs and adjuvant hypofractionated SRS found a durable local control of 71% at 3 years with grade 3 or higher neurologic toxicity of 4.1% more than 6 months after treatment, supporting effective local control with post-operative SRS [67]. Active areas of investigation include optimal fractionation as well as timing relative to surgery. The ongoing ALLIANCE trial A071801 (NCT04114981) is evaluating one or multiple fractions of SRS to the surgical bed. A phase II trial (NCT05871307 is investigating the treatment response and local tumor control of neoadjuvant SRS, intraoperative RT, or adjuvant SRS in the treatment of BrMs. In addition, three ongoing phase III trials are comparing local and distal brain recurrence, including the development of leptomeningeal disease, following pre-operative or post-operative SRS and surgery for BrMs (NCT03741673, NCT03750227, NCT05438212). In contrast to NSCLC, the data on SCLC regarding efficacy and timing are limited and, thus, surgery is reserved in these patients for the management of mass effect or symptoms [27].

### 2.4. Re-Irradiation

Given the increasing efficacy of targeted therapies and combinatorial strategies leading to increased survival, lung cancer patients are increasingly needing multiple courses of radiation for BrMs. The data on re-irradiation in the context of BrMs are limited, although initial clinical trials on SRS included WBRT followed by SRS or SRS followed by WBRT (Andrews et al., 2004; Aoyama et al., 2006; Brown et al., 2016; Chang et al., 2009; Kocher et al., 2011). SRS followed by WBRT or vice versa has been demonstrated to be safe, although WBRT has been associated with significant neurocognitive decline. Limitations of re-irradiation with SRS include the dose to organs at risk and the volume of normal brain tissue receiving certain dose levels. For SRS, the volume of brain tissue receiving 18 Gy (V18) has been shown to be predictive of radionecrosis and for V18 > 30.2 mm^2^, the rate of radionecrosis was 14% [51]. Additional studies are needed to further characterize the limitations of re-irradiation with SRS, although it appears to be safe within acceptable dose constraints.

**Table 1 cancers-16-03780-t001:** Key clinical trials defining radiation therapy for brain metastases.

Comparison	Trial Name or 1st Author	No. of BrM	Arms	No. of Patients	% Lung Cancer	Primary Endpoint	Additional Outcomes	Note	Citation
WBRT dose escalation	Borgelt et al.	Not specified	1st study - 30 Gy in 2 weeks - 30 Gy in 3 weeks - 40 Gy in 3 weeks - 40 Gy in 4 weeks 2nd study - 20 Gy in 1 week - 30 Gy in 2 weeks - 40 Gy in 3 weeks	1st study: n = 233 (30 Gy/2 wks) n = 217 (30 Gy/3 wks) n = 233 (40 Gy/3 wks) n = 227 (40 Gy/4 wks) 2nd study: n = 447 (20 Gy/1 wk) n = 228 (30 Gy/2 wks) n = 227 (40 Gy/3 wks)	60%	Overall rate of improvement in neurologic function 47% in 1st study, 52% in 2nd study	Median OS 18 weeks in the 1st study and 15 weeks in the 2nd study.	Patients with higher neurologic function at presentation had higher rate of improvement in neurologic function	[33]
WBRT vs. best supportive care	QUARTZ	Not specified	WBRT vs. optimal supportive care (OSC)	n = 269 (WBRT) n = 269 (OSC)	100%, NSCLC	46.4 vs. 41.7 days of QALYs (90% CI −12.7 to 3.3 days)	OS comparable between WBRT and OSC (HR 1.06, 95% CI 0.90–1.26, *p* = 0.8084)	20 Gy in 5 fractions lower than what is used at some institutions; improved OS with WBRT for age < 60.	[37]
WBRT + memantine	RTOG 0614	Not specified	WBRT + placebo vs. WBRT + memantine	n = 252 (placebo) n = 256 (memantine)	70%	Trend towards less decline in delayed recall (0 vs. −0.90, *p* = 0.059) with memantine relative to WBRT alone	Longer time to cognitive decline with memantine (HR 0.78; 95% CI 0.62–0.99, *p* = 0.01)		[38]
Hippocampal avoidance (HA) WBRT	NRG CC001	Not specified	WBRT + memantine vs. HA-WBRT + memantine	n = 257 (WBRT) n = 261 (HA-WBRT)	59%	Cognitive failure (measured as decline in reliable change index on at least one cognitive test) lower with HA-WBRT vs. WBRT (HR 0.74, 95% CI 0.58–0.95, *p* = 0.02)	Reduced deterioration in executive function at 4 months (40.4% vs. 23.3%, *p* = 0.01) and in learning/memory at 6 months (24.7% vs. 11.5%, *p* = 0.049); OS 7.6 vs. 6.3 months, *p* = 0.31; intracranial PFS 5.3 vs. 5.0 months, *p* = 0.21		[39]
WBRT +/− SRS	RTOG 9508	1–3	WBRT vs. WBRT + SRS	n = 164 (WBRT) n = 167 (WBRT + SRS)	64%	Median OS 6.5 vs. 5.7 months (*p* = 0.1356)	Improved control of treated lesions at 1 year in WBRT + SRS group (82% vs. 71% *p* = 0.01)		[18]
SRS +/− WBRT	Aoyama et al.	1–4	WBRT + SRS vs. SRS alone	n = 65 (WBRT + SRS) n = 67 (SRS alone)	67%	Medians OS 7.5 vs. 8.0 months (*p* = 0.42)	12-month brain tumor recurrence rate 46.8% vs. 76.4% (*p* < 0.001).	Salvage brain treatment less frequent in the WBRT + SRS group (n = 10) vs. SRS alone (n = 29) (*p* < 0.001); SCLC excluded	[20]
	Chang et al. (NCT00548756)	1–3	SRS +WBRT vs. SRS alone	n = 28 (SRS + WBRT) n = 30 (SRS alone)	55%	Mean probability of 5 point decline in HVLT-R-TR at 4 months 52% vs. 24% (96% confidence)	Median OS (actuarial) 5.7 vs. 15.2 months (*p* = 0.003) Local PFS (actuarial) at 4 months 73% vs. 27% (*p* = 0.0003)	Terminated early due to difference in primary outcome; SCLC excluded.	[21]
	EORTC 22952–26001	1–3	surgery or SRS -> WBRT vs. observation	n = 179 (SRS or surgery alone) n = 180 (SRS or surgery + WBRT)	53%	Median time to WHO performance status deterioration to more than 2 was 10.0 vs. 9.5 months (*p* = 0.89)	Median OS 10.9 vs. 10.7 months (*p* = 0.89). 2-year relapse rate at initial sites reduced with WBRT (surgery 59% vs. 27% *p* < 0.001; SRS 31 vs. 19%, *p* = 0.040)	SCLC excluded	[22]
	Alliance NCCTG N0574	1–3	SRS + WBRT vs. SRS alone	n = 102 (SRS + WBRT) n = 111 (SRS alone)	69%	Cognitive deterioration (decline > 1 standard deviation from baseline on.1 cognitive test) 91.7% vs. 63.5% (*p* < 0.001)	Quality of life −12.0 vs. −1.0 pts *p* = 0.001 at 3 months. Time to intracranial failure (HR 3.6, 95% CI 2.2–5.9, *p* < 0.001) shorter for SRS relative to WBRT. Median OS 7.4 vs. 10.4 months (*p* = 0.92)	SCLC excluded	[23]
SRS dose escalation & fractionation	RTOG 90-05	1	≤20 mm 18–24 Gy 21–30 mm 15–24 Gy 31–40 mm 12–18 Gy	n = 156 patients	43% NSCLC or breast	For ≤20 mm, grade 3–5 neurotoxicity in 11% of pts with 21 Gy and 10% with 24 Gy. For 21–30 mm, grade 3–5 neurotoxicity in 20% of pts with 18 Gy, 38% with 21 Gy, and 58% with 24 Gy. For 21–40 mm, grade 3–5 neurotoxicity in 14% of pts with 15 Gy, 50% with 18 Gy.	The actuarial radionecrosis incidence was 5%, 8%, 9%, and 11% at 6, 12, 18, and 24 months after SRS, respectively	For tumors < 20 mm, 24 Gy dose determined by investigator preference not to escalate to 27 Gy, rather than excessive toxicity	[53]
	Kim et al.	1–3	24 Gy in 3 fractions 27 Gy in 3 fractions 30 Gy in 3 fractions	n = 15 (24 Gy/3 fx) n = 17 (27 Gy/3 fx) n = 14 (30 Gy/3 fx)	72%	The 6-month cumulative incidence of radiation necrosis was 0% with 24 Gy/3 fx, 13% (95% CI 0–29%) with 27 Gy/3 fx, and 37% (95% CI 1–58%) with 30 Gy/3 fx.	12-month PFS 65% with 24 Gy/3 fx, 80% with 27 Gy/3 fx, and 75% with 30 Gy/3 fx. 24 Gy/3 fx significantly associated with local treatment failure (*p* = 0.037)		[52]
	NRG-BN013	1–8	SRS in 1 fraction vs. SRS in 3 fractions	n = 269 (estimated enrollment)	Not yet reported	Time to local failure; not yet reported	Intracranial PFS, overall survival, differences in failure patterns, rates of radiation necrosis, time to salvage WBRT, post-treatment adverse events	maximum diameter ≥ 1.0 cm and ≤3.0 cm	NA
SRS to an increasing number of lesions	Yamamoto (JLGK0901)	1–10	NA	n = 455 (1 BrM) n = 531 (2–4 BrM) n = 208 (5–10 BrM)	76% (included SCLC)	Non-inferior OS for 2–4 tumors vs. 5–10 HR 0.97, 95% CI 0.81–1.18, *p* < 0.0001 for non-inferiority	Proportion of SRS-induced grade 3–4 adverse events: 2% of pts with 1 tumor, 2% of pts with 2–4 tumors, and 3% of pts with 5–10 tumors		[25]
	Li et al. (NCT01592968)	4–15	SRS vs. WBRT	n = 50 (SRS) n = 50 (WBRT)	Not yet reported	HVLT_R_TR (memory function) at 4 months Z-score decreased by 0.21 vs. 0.74 (*p* = 0.041) Local control at 4 months 95% vs. 87% (*p* = 0.79)	Median OS 7.8 vs. 8.9 months (*p* = 0.59) Time to systemic therapy 1.7 vs. 4.1 weeks (*p* = 0.001)	Early termination due to NRG CC001	[55]
	NCT03075072	5–20	SRS vs. HA-WBRT	n = 196 (estimated enrollment)	Not yet reported	Quality of life (MDASI-BT)	Overall survival, neurologic survival, incidence/time to detection of new BrM, incidence/time to LR of treated BrM, development of radionecrosis, development of LMD, incidence/time to salvage surgery or WBRT, time to neurologic decline	SCLC excluded	NA
	NCT03550391	5–15	SRS vs. HA-WBRT + memantine	n = 206 (estimated enrollment)	Not yet reported	Overall survival, neurocognitive PFS	Time to CNS failure, differences in CNS failure patterns, number of salvage procedures, time to (re)initiation of systemic therapy, estimated cost, quality of life	SCLC excluded; Will collect plasma/serum samples, imaging features on MRI that may predict tumor control/neurocognitive outcomes, and dosimetry	NA
	NCT04277403 (HIPSTER_2020)	4–15	SRS vs. HA-WBRT + simultaneous boost	n = 150 (estimated enrollment)	Not yet reported	Intracranial progression free survival	Neurocognitive function (VLMT, COWAT, TMT), local control rate (MRI/FET-PET), survival, quality of life	SCLC excluded; simuntaneously integrated boost to each metastasis of 51 Gy to 95% of PTV in 12 fractions 4.25 Gy/fraction	NA
	NCT03775330	5–30	SRS vs. SRS + WBRT	n = 126 (estimated enrollment)	Not yet reported	HLVT-R-TR 2 months post treatment	Additional neurocognitive tests including HLVT-R, TMT, COWA, and CTB COMP; local control, distant brain failure, overall CNS response, overall survival, and quality of life	SCLC excluded	NA
SRS + tumor treating fields (TTF)	METIS (EF-25) (NCT02831959)	1–10	SRS vs. SRS + TTF	n = 298	100%, NSCLC	Time to first intracranial progression 11.3 vs. 21.9 months with the addition of TTF (HR 0.67, 95% CI 0.48–0.93, *p* = 0.02)	Adverse events with TTF mainly dermatological; TTF	no prior WBRT	[59]
Surgery + post-op RT	“Patchell I”	1	Surgery + WBRT vs. WBRT alone	n = 25 (surgery +WBRT), n = 23 (WBRT)	77%	Median OS 40 vs. 15 weeks LC 80% vs. 48%	Functional independence 38 vs. 8 weeks *p* < 0.005		[16]
	“Patchell II”	1	Surgery + WBRT vs. Surgery + observation	n = 49 (WBRT), n = 46 (obs.)	60%	Intracranial recurrence 18% vs. 70% *p* < 0.001	DBC 86% vs. 63% *p* < 0.01 LC 90% vs. 54% *p* < 0.001 Neurologic death 14% vs. 44% *p* = 0.003 No difference in OS	RT dose 50.4 Gy/28 fractions higher than standard of care	[17]
	Mahajan et al. (NCT00950001)	1–3	Surgery + SRS vs. Surgery + observation	n = 64 (SRS) n = 68 (observation)	10%	12-month local tumor free recurrence rate was 72% (95% CI 60–87%) vs. 43% (95% CI 31–59%) with a HR 0.46 (95% CI 0.24–0.88, *p* = 0.015)	Median OS 17 vs. 18 months (*p* = 0.24). The probability of being free of distant BrM at 12 months was 42% vs. 33% (*p* = 0.35)	46% of patients in the observation arm underwent WBRT compared to 38% in the SRS arm	[64]
	NCCTG N107C/CEC·3	1–4	SRS to cavity vs. WBRT	n = 98 (SRS), n = 96 (WBRT)	59%	Cognitive-deterioration free survival 3.7 vs. 3.0 mo. *p* < 0.0001 OS 12.2 vs. 11.6 months *p* = 0.70	LC 61.8% vs. 87.1% at 1 y *p* = 0.00016 DBC 64.7% vs. 89.2% at 1 y *p* = 0.00045	After post-hoc central review, no diffference in LC between SRS and WBRT	[65,66]
SRS vs. WBRT for SCLC	ENCEPHALON	1–10	SRS or SRT (for >3 cm) vs. WBRT	n = 26 (SRS or SRT) n = 25 (WBRT)	100%, SCLC	Drop of at least 5 points from baseline in HVLT-R 7.7% vs. 24% (*p* = 0.0723)	median OS 124 vs. 131 days (*p* = 0.36), QoL	inclusion criteria included KPS > 50, which is lower than trials for NSCLC	[62]
	Ayal Aizer et al. (NCT03391362)	1–10	SRS or SRT (for >2 cm) vs. WBRT	n = 66 (SRS) n = 32 (SRT to at least 1 site)	100%, SCLC	Rate of neurologic death at 1 year 11.0% (95% CI 4.8–17.2%)	Median OS 10.3 months	Compared to their WBRT outcomes indirectly, which showed 1-year rate of neurologic death in 17.5% of pts	[63]
	NCT06457906 (SHARP)	1–10	SRS or SRS vs. HA-WBRT	n = 340 (estimated enrollment)	100%, SCLC	Cognitive function (HVLT-R-TR at 6 months), median overall survival	MoCA score at 3, 6, 9, and 12 months, quality of life scores, neurological PFS, intracranial PFS, PFS, treatment-related toxicities		NA
	NRG CC009 (NCT04804644)	1–10	SRS vs. HA-WBRT	n = 200 (estimated enrollment)	100%, SCLC	Cognitive function failure (cognitive decline measured on HVLT-R, COWA, or TMT)	Perceived difficulties in cognitive abilities, symptom burden (MDASI-BT), intracranial PFS, overall survival, incidence of neurologic death, adverse effects, radionecrosis		NA
PCI	Auperin et al. (meta-analysis of 7 trials)	NA	PCI vs. observation	n = 987	100%, SCLC	OS at 3 years 20.2% with PCI vs. 15.3% without (HR 0.84, 95% CI 0.73–0.97, *p* = 0.01).	Decreased incidence of BrM with PCI (RR 0.46, 95% CI 0.38–0.57, *p* < 0.001)		[32]
	Slotman	NA	PCI vs. observation	n = 143 (PCI) n = 143 (observation)	100%, SCLC	PCI associated with decreased risk of symptomatic BrM (HR 0.27, 95% CI 0.16–0.44, *p* < 0.001)	Median DFS 12.0 vs. 14.7 weeks (*p* = 0.02), median OS 5.4 vs. 6.7 months (*p* = 0.003)		[33]
	Takahashi et al.	NA	MRI surveillance vs. PCI	n = 111 (observation) n = 113 (PCI)	100%, SCLC (ES)	Median overall survival 11.6 vs. 13.7 months (*p* = 0.094)	Most frequent grade 3 or worse adverse events were anorexia (2% vs. 6%), malaise (1% vs. 3%), and weakness in a lower limb (5% vs. 1%)	Terminated at interim analysis due to PCI being superior to observation	[46]
	MAVERICK (SWOG S1827)	NA	MRI surveillance vs. PCI	n = 668 (estimated enrollment)	100%, SCLC (LS & ES)	Overall survival; not yet reported	Cognitive failure free survival, brain-metastasis-free survival, frequency/severity of toxicities		NA
	EORTC PRIMALung	NA	MRI surveillance vs. PCI	n = 600 (estimated enrollment)	100%, SCLC (LS & ES)	Overall survival; not yet reported	Cognitive failure-free survival, quality of life, and acute/late toxicities		[47]
HA-PCI	PREMER	NA	PCI vs. HA-PCI	n = 75 (PCI) n = 75 (HA-PCI)	100%, SCLC (LS & ES)	Decline in delayed free recall on the FCSRT at 3 months was 23.5% vs. 5.8% *p* = 0.003	Quality of life, incidence of BrM, and OS not significantly different between the 2 groups		[43]
	Belderos et al. (NCT01780675)	NA	PCI vs. HA-PCI	n = 84 (PCI) n = 84 (HA-PCI)	100%, SCLC (LS & ES)	HVLT-R-TR decrease of 5 points or more in 29% (PCI) vs. 28% (HA-PCI) *p* = 1.00.	No significant differences on any cognitive tests between the groups Cumulative incidence of BrM at 2 years 20% (95% CI 12–29%) vs. 16% (95% CI 7–24%) *p* = 0.60 Median OS 19.9 vs. 18.5 months *p* = 0.70		[44]
	NRG CC003	NA	PCI vs. HA-PCI	n = 392 patients total	100%, SCLC	Non-inferior 12-month intracranial response rate (14.8% PCI vs. 14.2% HA-PCI, *p* < 0.0001 non-inferiority) 6 month HVLT-R-DR deterioration not significant 30% vs. 26%, *p* = 0.31	No significant difference in OS (adjusted HR 0.83, 95% CI 0.63–1.09, *p* = 0.19)	HA-PCI arm had lower COWA scores at bseline and showed greater deterioration in COWA over time (−0.259, *p* = 0.042)	[45]

## 3. Systemic Therapy

### 3.1. Targeted Therapies

In NSCLC, especially adenocarcinoma, personalized therapeutic approaches have led to the development of increasingly efficacious targeted inhibitors for patients with actionable alterations, including in *EGFR*, *ALK*, *KRAS*, *BRAF*, *RET*, *ROS1*, *MET*, and *NTRK* [68]. There are increasing data for the CNS efficacy of newer-generation tyrosine kinase inhibitors (TKIs), although much of the data still come from secondary analyses of subsets of prospective trials (Table 2).

In NSCLC patients with canonical *EGFR* mutations, osimertinib is a third-generation TKI developed to overcome resistance to the *EGFR* T790M mutation [69], shown to exhibit penetrance of the blood–brain barrier in non-human primate studies [70]. In pre-planned analyses of patients with measurable and/or nonmeasurable CNS lesions in the phase III trials AURA3 [71] and FLAURA [72], osimertinib monotherapy led to a 40% CNS ORR in patients with *EGFR* T790M and resistance to previous TKI treatment and a 66% CNS ORR in the first-line setting; higher CNS ORRs were seen in patients with ≥1 measurable CNS lesion (Table 2). A phase II cohort study of 160 mg osimertinib NSCLC patients with *EGFR* T790M and BrMs or LM demonstrated an intracranial ORR of 55% in the BrM patients and intracranial progression was observed in 18.5% of the LM patients [73], suggesting that doubling the dose could be a reasonable strategy for the local control of BrMs and LM. In patients without CNS metastases, the ADAURA study of 682 stage IB-IIIA NSCLC patients showed that the CNS DFS HR was 0.36 (95% CI 0.23 to 0.57) for adjuvant osimertinib relative to placebo, suggesting a role for osimertinib in CNS prophylaxis [74]. In contrast to typical *EGFR* mutations, data on the intracranial efficacy of inhibitors targeting *EGFR/ERBB2* exon 20 insertions in NSCLC have been limited [75,76] (Table 2).

Second- and third-generation ALK inhibitors have also shown promising intracranial control in patients with *ALK* fusion-positive NSCLC [77,78,79,80,81,82] (Table 2). A post hoc analysis of the phase III ALEX trial randomizing treatment-naïve patients with *ALK* mutations to alectinib or crizotinib demonstrated that the intracranial ORR was 85.7% with alectinib (n = 25) vs. 71.4% with crizotinib (n = 21) in patients who had received prior radiotherapy and 78.6% (n = 21) versus 40.0% (n = 22) in patients with previously untreated BrMs [78]. In addition, the CROWN trial randomized 296 treatment-naïve patients with *ALK*-mutated NSCLC to lorlatinib or crizotinib, and a post hoc analysis demonstrated that in the patients with baseline BrMs (n = 37 lorlatinib, n = 39 crizotinib), the intracranial ORR was 65% vs. 18% favoring lorlatinib [81]. Importantly, the median time to intracranial progression with lorlatinib was not reached versus 16.4 months with crizotinib, suggesting durable responses were achieved with lorlatinib. Moreover, a phase II trial evaluated lorlatinib in patients who had previously received one to three prior ALK TKIs; among 57 patients with BrMs at baseline, the intracranial ORR was 56.1% [77].

In the Ras-Raf pathway, inhibitors of *KRAS* G12C have demonstrated modest local control for intracranial metastasis, while data on *BRAF* V600E in NSCLC patients are limited [83,84] (Table 2). The phase II CodeBreaK100 trial demonstrated an ORR of 37.1% and a median duration of response (DoR) of 11.1 months for sotorasib in *KRAS* G12C-mutated NSCLC, although intracranial outcomes were not assessed [85]. Adagrasib exhibited an intracranial ORR of 33.3% (94% CI 18.0–51.8) among 33 patients with NSCLC with *KRAS* G12C mutations with previously treated stable BrMs [86]. In the phase I/II trial of garsorasib in *KRAS* G12C-mutated NSCLC, 11 of 79 patients had stable BrMs and exhibited an intracranial ORR of 17% [87].

*RET*, *ROS1*, *MET*, and *NTRK* encode receptor tyrosine kinases that are mutated in a small proportion of lung adenocarcinomad and for which selective and multi-kinase inhibitors have recently been developed with CNS activity (Table 2). In the phase I/II ARROW, LIBRETTO-001, and LIBRETTO-321 trials assessing selective RET inhibitors in patients with *RET* fusion-positive NSCLC, a small portion of patients had measurable BrMs at baseline and exhibited intracranial response rates of 56–91% [88,89,90]. In addition, a post hoc analysis of the LIBRETTO-001 and LIBRETTO-201 trials of selpercatinib in *RET* fusion-positive NSCLC found that the incidence of BrMs remained at 0% for up to 36 months in 30 patients without BrMs at baseline [91].

Similarly, potent and selective ROS1 inhibitors were shown to have high rates of intracranial ORR (79–89%) in phase I/II clinical trials in patients with NSCLC positive for *ROS* fusions [92,93] (Table 2). In patients with *MET* exon 14 skipping mutations, which act as oncogenic drivers in 3–4% of NSCLC [68], selective MET inhibitors were found to cause intracranial responses in 7 of 13 (53.8%) and 5 of 7 (71.4%) patients with measurable BrMs at baseline in the GEOMETRY mono-1 ad VISION trials, respectively [94,95]. In patients with *NRTK1/2/3* gene fusions, the tropomyosin receptor kinase inhibitor entrectinib demonstrated an intracranial response in 6 of 11 patients with *NRTK1/2/3* fusion-positive NSCLC and BrMs at baseline [91].

The antibody–drug conjugate trastuzumab-deruxtecan (T-DXd) has demonstrated efficacy in *HER2*-mutated NSCLC metastatic to the brain. In the phase II DESTINY-Lung-01 trial of T-DXd in unresectable and/or metastatic HER2-expressing NSCLC, the *HER2*-mutated cohort included 33 patients with stable BrMs and showed an intracranial ORR of 54.5% [96], while a intracranial response was not reported in the HER2-overexpressng cohort [97]. Similarly, the phase II DESTINY-Lung-02 trial evaluating the dosing of T-DXd in *HER2*-mutated metastatic NSCLC demonstrated CNS response rates of 45–60% [98]. In patients with *EGFR*-mutated NSCLC, the HERTHENA-Lung01 phase II trial evaluated patritumab deruxtecan (HER3-DXd) in patients who progressed with EGFR TKI therapy and platinum chemotherapy. In 30 patients with baseline BrMs (no prior radiotherapy), the intracranial ORR was 33.3% (95% CI 17.3–52.8) [99]. The ongoing HERTHENA-Lung02 phase III trial is randomizing patients with progression on third-generation EGFR TKIs to HER3-DXd or platinum chemotherapy and will evaluate the intracranial PFS along with secondary endpoints (NCT05338970) [100].

Bispecific antibodies can inhibit ligand binding for signaling pathways as well as lead to the activation of the anti-tumor immune response and increasingly are being tested in lung cancer [101]. The PAPILLON trial examined the addition of amivantamab, a bispecific antibody targeting EGFR and c-MET that can activate monocytes/macrophages and NK cells through its Fc domain, to chemotherapy in patients with NSCLC with *EGFR* exon 20 insertions [102]. Although the addition of amivantamab led to an ORR of 73% compared to 47% with chemotherapy alone, the CNS efficacy was not evaluated and patients with untreated BrMs were excluded. In the phase III MARIPOSA trial, patients with *EGFR*-mutated NSCLC were randomized to the first-line treatment of osimertinib or amivantamab with lazertinib, which demonstrated a benefit on the PFS in patients with a history of BrMs with amivantamab and lazertinib (HR 0.69, 95% CI 0.53–0.92), although the intracranial ORR and PFS were not described [103]. The MARIPOSA-2 trial examined the intracranial response rates in patients with *EGFR*-mutated NSCLC treated with amivantamab with carboplatin-pemetrexed with or without lazertinib and found that amivantamab with chemotherapy led to prolonged median intracranial PFS relative to chemotherapy alone (12.5 versus 8.3 months) [104]. The majority of trials of bispecific antibodies in lung cancer to date have excluded symptomatic and/or active BrMs [102,103,104,105,106,107,108,109,110,111,112,113,114,115,116,117,118] (Table 3). Randomized studies evaluating the impact on CNS efficacy and prevention of BrMs of bispecific antibodies are needed to elucidate if they are effective in achieving an intracranial response or preventing the formation of new BrMs.

Collectively, the findings from these trials demonstrate that targeted inhibitors with CNS penetrance offer the potential for CNS disease control in NSCLC. Further randomized trials with an emphasis on the degree and duration of the intracranial response and long-term CNS PFS are needed to evaluate the durability of CNS responses in NSCLC with targetable mutations and BrMs. The combination of targeted inhibitors with platinum chemotherapy is also being evaluated as a strategy for achieving higher rates of local control for BrMs and preventing the development of new CNS lesions. The GAP-BRAIN trial randomized patients with untreated *EGFR*-mutant NSCLC BrMs to gefitinib alone or gefitinib with pemetrexed and platinum chemotherapy [119]. Gefinitib and chemotherapy improved the intracranial PFS (15.6 months (95% CI, 14.3–16.9 months) vs. 9.1 months (95% CI, 8.0–10.2 months)) compared with gefitinib alone. The intracranial ORR was 85% with gefinitib and chemotherapy vs. 63% with gefitinib alone (*p* = 0.002), although grade 3 and higher adverse events were common [119]. The phase III FLAURA2 trial randomized patients to osimertinib monotherapy or osimertinib with platinum-pemetrexed and included brain scans of all patients at baseline and progression [120]. The CNS ORR was 73% in the combination-treated versus 69% in the monotherapy group. The intracranial PFS was 20.1 (0–33.3) months in the combination arm and 13.9 (0–33.1) months in the monotherapy arm. A critical limitation of osimertinib with platinum chemotherapy was a doubling of the incidence of grade 3–5 adverse events as well as a doubling of fatal events in the combination treatment arm compared to those with osimertinib alone [120,121].

### 3.2. Chemotherapy and Immunotherapy

First-line platinum-based chemotherapy has been shown to yield an intracranial ORR of 20–30% in patients with stage IV NSCLC with BrMs [122] and cisplatin and pemetrexed can achieve CNS response rates up to 40% in lung adenocarcinoma [123]. Immune checkpoint inhibitors (ICIs) are an established strategy in the management of metastatic NSCLC, especially in patients without oncogene driver mutations (NCCN guidelines). However, data on the efficacy of ICIs in BrMs are limited. A prospective study of NSCLC patients treated with programmed cell death 1/programmed cell death ligand 1 (PD-1/PD-L1) blockade with or without anti–cytotoxic T-lymphocyte antigen 4 (CTLA4) included 255 patients with BrMs, of which 39.2% were active, 14.3% were symptomatic, and 27.4% were treated with steroids, demonstrating an intracranial ORR of 27.3% in the active BrMs [124]. In a phase 2 trial evaluating pembrolizumab in patients with NSCLC or melanoma and untreated BrMs (5–20 mm in size, not previously treated or with progression after prior radiotherapy), intracranial responses were observed in 29.7% of the cohort with at least 1% PD-L1 expression (n = 37); no intracranial responses occurred in the patients with less than 1% or unevaluable PD-L1 expression (n = 5) [125]. A post hoc analysis of the Checkmate 227 trial part 1 examined intracranial outcomes in NSCLC patients with tumor PD-L1 expression 1% or greater treated with nivolumab plus ipilimumab or chemotherapy [126]. The intracranial progression-free survival at 5 years was 16% (95% CI: 5–33) with nivolumab plus ipilimumab and 6% (95% CI: 1–22) with chemotherapy. In the patients with baseline BrMs, 4% developed new brain lesions with nivolumab plus ipilimumab compared to 20% with chemotherapy, while the rates of new BrMs in the patients without baseline BrMs were comparable [126].

Although the blood–brain barrier [127] and relatively immune-privileged status of the CNS [128] were thought to limit the access of systemic therapies to the brain, these studies suggested that chemotherapy and ICIs have some intracranial efficacy against BrMs. Given the low intracranial response rates to either modality, ongoing trials are evaluating the efficacy of combining PD-1/PD-L1 blockade and chemotherapy in treatment-naïve NSCLC with BrMs. The phase II CAP-BRAIN trial has revealed an intracranial ORR of 52.5% in treatment-naïve patients with nonsquamous NSCLC without *EGFR* mutations or *ALK* translocations and at least three BrMs treated with camrelizumab and carboplatin/pemetrexed [129]. Similarly, the phase II Atezo-Brain trial evaluated atezolizumab with carboplatin/pemetrexed in *EGFR*- and *ALK*-negative nonsquamous NSCLC with untreated asymptomatic or symptomatically controlled BrMs [130]. The median intracranial PFS was 6.9 months and the intracranial ORR was 42.7%; an exploratory analysis showed a comparable intracranial response in the patients receiving steroids (50%) compared to those who were not (39%), suggesting corticosteroid use did not impair the intracranial responses in the context of ICIs and chemotherapy in this trial [130].

Targeting the signaling of vascular endothelial growth factor (VEGF) has also been proposed to prevent BrM formation in lung adenocarcinoma [131]. A phase II trial examined ramucirumab (anti-VEGFR-2) and docetaxel in stage IV NSCLC after progression on prior chemotherapy [132]. In 25 patients, the median intracranial PFS was 4.6 months and the intracranial ORR was 20%. The phase III Impower150 study randomized patients with chemotherapy-naïve stage IV nonsquamous NSCLC to atezolizumab plus carboplatin/paclitaxel (ACP), bevacizumab (anti-VEGF) plus carboplatin/paclitaxel (BCP), or atezolizumab plus bevacizumab plus carboplatin/paclitaxel (ABCP) [133]. Although patients with untreated BrMs were excluded, CNS imaging was performed at baseline and as clinically indicated. New BrMs developed in 11.9% of the patients treated with ACP, 6.0% of the patients treated with BCP, and 7.0% of the patients treated with ABCP; the median time to development of new BrMs was not met. Randomized studies examining VEGF/VEGFR inhibition in patients with BrMs are needed to further evaluate if targeting this pathway is efficacious in preventing the development of BrMs in nonsquamous NSCLC [133].

In SCLC, objective intracranial responses to chemotherapy have been reported in the range of 30%, including in the case of symptomatic BrMs [134,135]. In a large retrospective study, first-line combination chemotherapy achieved a systemic response rate of 73% and an intracranial response rate of 27%; however, the time to symptom progression was comparable between the intracranial responders and non-responders [136]. In a randomized trial from the EORTC, the addition of WBRT to teniposide increased the intracranial response rate to 57% [137]. In these patients, the survival was poor (the median was approximately 3 months) and WBRT offered no significant improvement in survival over teniposide therapy alone.

Immune checkpoint blockade was shown to produce systemic response rates up to 33% in patients with ES-SCLC [138,139]. Recent clinical trials have shown CNS efficacy in SCLC with the addition of PD-1/PD-L1 blockade to chemotherapy. The IMPower133 trial evaluated the addition of atezolizumab to carboplatin and etoposide, with 11% of patients receiving PCI in both arms. In a secondary analysis of the trial, atezolizumab led to a doubling of the time to intracranial progression in both patients who had received PCI and in those who had not [140]. In addition, the bispecific antibody tarlatamab targeting delta-like ligand 3 (DLL3) and the CD3 T-cell co-receptor has shown systemic efficacy in previously treated SCLC; patients with stable, treated BrMs were allowed, although the intracranial efficacy outcomes have not been reported [141].

**Table 2 cancers-16-03780-t002:** Intracranial efficacy of selected targeted therapies in lung cancer.

Drug Class	Target(s)	Trial Name	Arms	Phase	Patient Characteristics	No. of Pts with BrM	CNS Disease Inclusion Criteria	Measurement of CNS Response	CNS Outcomes	Reference
Targeted inhibitor (e.g., tyrosine kinase inhibitors, inhibitors of MAPK pathway)	Typical EGFR (L858R, Ex19del, T790M)	AURA3 (NCT02151981)	Osimertinib vs chemotherapy	III	Progression on 1st line EGFR TKI treatment and positive T790M mutation	Any BrM: - n = 75 (osimertinib) - n = 41 (chemo.) Measurable BrM: - n = 30 (osimertinib) - n = 16 (chemo.)	Asymptomatic, stable CNS metastases not requiring steroids at least 4 weeks before study entry	RECIST 1.1 RANO-LM for LMD	CNS ORR - any BrM 40% (osimertinib) vs. 17% (chemo) *p* = 0.014 - measurable BrM 70% (osimertinib) vs. 31% (chemo) *p* = 0.015 CNS median PFS: 11.7 (osimertinib) vs. 5.6 months (chemo) *p* = 0.004	[71]
		FLAURA (NCT02296125)	Osimertinib vs. standard EGFR-TKI	III	Stage IIIB-IV Ex19del or L858R Treatment-naïve	Any BrM: - n = 61 (osimertinib) - n = 67 (standard TKI) Measurable BrM: - n = 22 (osimertinib) - n = 19 (standard TKI)	Asymptomatic, stable CNS metastases included; if symptomatic, neurologic status stable ≥2 weeks after completion of definitive therapy and steroids	RECIST 1.1	CNS ORR - any BrM 66% (osimertinib) vs. 43% (TKI) *p* = 0.011 - measurable BrM 91% (osimertinib) vs. 68% (TKI) *p* = 0.066 CNS median PFS (any BrM): NR (osimertinib) vs. 13.9 months (TKI) *p* = 0.014	[72]
		ADAURA (NCT02511106)	Osimertinib vs Placebo	III	Stage IB-IIIA NSCLC Ex19del or L858R After complete tumor resection	Not reported	Not specified	NA	CNS recurrence in 6% (osimertinib) vs. 11% (placebo) CNS DFS HR not reached in osimertinib or placebo group in Stage II-IIIA, CNS DFS HR 0.24 (95% CI 0.14–0.42) favoring osimertinib	[74]
		FLAURA 2 (NCT04035486)	Osimertinib vs. Osimertinib + chemotherapy	III	Stage IIIB-IV Ex19del or L858R Treatment-naïve	Any BrM: - n = 104 (osimertinib) - n = 118 (osi. + chemo.) Measurable BrM: - n = 38 (osimertinib) - n = 40 (osi. + chemo.)	Asymptomatic, stable CNS metastases included; if symptomatic, neurologic status stable ≥2 weeks after completion of definitive therapy and steroids	RECIST 1.1	CNS ORR - any BrM 69% (osimertinib) vs. 73% (combination) *p* = 0.55 (43% vs. 59% CR) - measurable BrM 87% (osimertinib) vs. 88% (combination) *p* = 0.93 CNS median PFS (any BrM): NR (osimertinib) vs. 13.9 months (TKI) *p* = 0.014	[120]
		GAP-BRAIN (NCT01951469)	Gefitinib vs. Gefitinib + chemotherapy	III	Stage IV Ex19del or L858R Treatment-naïve	n = 81 (gefitinib) n = 80 (gefitinib + chemotherapy)	At least 3 BrM or patients with 1–2 intracranial lesions who were not suitable for or refused radiotherapy	Modified RECIST 1.1	CNS ORR 63% (95% CI 52–74%) vs. 85% (95% CI 77–93%) *p* = 0.002 Median CNS PFS 9.1 months (95% CI 8.0–10.2) vs. 15.6 months (95% CI 14.3–16.9) *p* < 0.001	[105]
	EGFR/ERBB2 Ex20ins	ZENITH20	Poziotinib	II	Stage IIIB-IV *HER2* exon 20 insertion or EGFR exon 20 insertion Cohort 1 & 2: previously treated Cohort 3: treatment-naïve	n = 36 (*EGFR* Ex20 insertion) n = 14 (*HER2* Ex20 insertion)	BrM allowed if asymptomatic or previously treated (must be stable on post-radiation MRI)	Modified RECIST 1.1	CNS ORR 8.3% (CR) CNS stable disease in 66.7%	[75]
		NCT02716116	Mobocertinib	I/II	Stage IIIb-IV NSCLC or other solid tumors Previously treated	n = 12	Active BrM allowed	NA	CNS ORR not reported Overall ORR 25% in patients with baseline	[76]
	ALK	NCT01970865	Lorlatinib	II	Stage IV NSCLC *ALK*+ (EXP3B-5) Previously treated with 2nd generation ALK TKI	n = 95 (n = 67 with prior brain-directed RT)	Asymptomatic treated or untreated CNS metastases were allowed; Excluded if SRS or partial brain RT within 2 weeks prior to randomization or WBRT within 4 weeks prior to randomization	Modified RECIST 1.1	Intracranial ORR 56.1% (95% CI 42.4–69.3%); median intracranial duration of response 12.4 months (95% CI 6.0–37.1)	[77]
		ALEX (NCT02075840)	Alectinib vs. crizotinib	III	Stage IIIB-IV *ALK*+ NSCLC (by IHC) Treatment naïve	Any BrM: - n = 25 (alectinib) - n = 21 (crizotinib)	BrM or LMD allowed if asymptomatic	RECIST 1.1 RANO-BM	CNS ORR - any BrM: 36% (alectinib) vs. 29% (crizotinib) with prior RT; 74% vs. 25% without prior RT - measurable BrM:86% vs. 71% with prior RT; 79% vs. 40% without prior RT 12 month cumulative incidence of CNS progression - BrM at baseline: 16% vs. 58% - No BrM at baseline: 4.6% vs. 31.5%	[78]
		J-ALEX	Alectinib vs. crizotinib	III	Stage IIIB-IV ALK+ NSCLC (by IHC) No prior TKI, no more than 1 regimen of systemic anticancer therapy	Any BrM: - n = 14 (alectinib) - n = 29 (crizotinib)	BrM or LMD allowed if asymptomatic (>2 weeks since last dose of steroids prior to enrollment)	RECIST 1.1	CNS progression longer with alectinib than crizotinib (HR 0.22, 95% CI 0.10–0.48, *p* < 0.0001)	[79]
		ALTA-1L	Brigatinib vs. crizotinib	III	Stage IIIB-IV ALK+ NSCLC (by IHC) Treatment naïve Stratified by presence of BrM	Any BrM: - n = 47 (brigatinib) - n = 49 (crizotinib) Measurable BrM: - n = 18 (brigatinib) - n = 23 (crizotinib)	Asymptomatic or stable BrM allowed (no increased dose of steroids or anticonvulsants for 7 d prior to randomization)	RECIST 1.1	Measurable BrM: CNS ORR 78% (95% CI 52–94%) vs. 26% (95% CI 10–48%) Any BrM: CNS ORR 66% (95% CI 51–79%) vs. 16% (95% CI 7–30%); CNS PFS at 2 years 48% vs. 15%, *p* < 0.0001	[80]
		CROWN (NCT03052608)	Lorlatinib vs. crizotinib	III	Stage IIIB-IV ALK+ NSCLC (by IHC) Treatment naïve	Baseline BrM: - n = 37 (lorlatinib) - n = 39 (crizotinib)	BrM allowed if asymptomatic and not currently requiring steroid treatment; Excluded if SRS or partial brain RT within 2 weeks prior to randomization or WBRT within 4 weeks prior to randomization	Modified RECIST 1.1	Measurable BrM: CNS ORR 82% vs. 23% Any BrM: CNS ORR 66% vs. 20%; CNS PFS improved with lorlatinib (HR 0.10, 95% CI 0.04–0.27) Pts without BrM: HR 0.02, 95% CI 0.002–0.14	[81]
		NCT04009317	Envonalkib vs. crizotinib	III	Stage IIIB-IV ALK+ NSCLC (by IHC) Treatment naïve	Baseline BrM: - n = 19 (envonalkib) - n = 21 (crizotinib)	BrM allowed if asymptomatic; patients stratified by presence of BrM at baseline	RECIST 1.1	CNS ORR 79% (95% CIwith envonalkib, 5% with crizotinib	[82]
	BRAF V600E	NCT01336634	Dabrafenib + trametinib	II	BRAF V600E NSCLC Cohort B: previously treated Cohort C: treatment-naïve	Cohort C: n = 2 (nonmeasurable, previously treated)	Asymptomatic, untreated BrM (<1 cm in the largest dimension) or treated BrM stable for 3 weeks allowed	RECIST 1.1	Both patients had non-CR and non-PD	[83]
		NCT03915951	Encorafenib + binimetinib	II	BRAF V600E NSCLC Treatment naïve or have received 1 line of platinum chemotherapy or anti-PD-1 + chemotherapy	n = 4 (treatment-naïve) n = 4 (prior treatment)	Excluded if symptomatic or active BrM or LMD	RECIST 1.1	One patient from each group experienced intracranial progression.	[84]
	KRAS G12C	CodeBreak100	sotorasib	II	Stage I-IV *KRAS G12C* NSCLC Previously treated	n = 26	BrM excluded if active and/or untreated	NA	Not reported	[85]
		KRYSTAL-1	adagrasib	I/II	Stage III-IV NSCLC *KRAS* G12C by PCR or NGS Previously treated	n = 42	BrM allowed if adequately treated and neurologically stable	NA	CNS ORR 33.3% (95% CI 18–51.8%); median CNS PFS 5.4 months	[86]
		NCT05383898	garsorasib	I	Stage III-IV NSCLC *KRAS* G12C by PCR or NGS	n = 11 (n = 6 with measurable BrM at baseline)	BrM excluded if unstable or progressive	RECIST 1.1	CNS ORR 17%; CNS disease control rate of 100% (note: median overall follow-up 8.8 months)	[87]
	RET	ARROW	pralsetinib	I/II	Treatment-naïve and previously treated	n = 9 (measurable BrM)	Excluded if BrM associated with progressive neurologic symptoms	RECIST 1.1	CNS ORR 56% (5 of 9 pts had CR)	[88]
		LIBRETTO-001 LIBRETTO-201	selpercatinib	I/II	Treatment-naïve and previously treated	n = 22 (measurable BrM)	Allowed if neurologically stable with stable steroid dose for 14 days before enrollment; no surgery or RT w/in 28 days and no SRS within 14 days allowed	RECIST 1.1	CNS ORR 82% (86% without prior RT, 75% with prior RT)	[89,91]
		LIBRETTO-321	selpercatinib	II	Treatment-naïve and previously treated	n = 5 (measurable BrM)	Allowed if asymptomatic or previously treated and stable disease for ≥2 weeks	RECIST 1.1	CNS ORR 80%	[90]
	ROS1	ALKA-371-001 STARTRK-1 STARTRK-2	entrectinib	I/II	Treatment-naïve and previously treated	n = 46 (any BrM) n = 24 (measurable BrM)	Allowed if asymptomatic or previously treated and stable disease	RECIST 1.1	Measurable BrM: CNS ORR 79.2%, CNS median PFS 12.0 months Any BrM: CNS ORR 52.2%, CNS median PFS 8.3 months	[92]
		TRIDENT-1	repotrectinib	II	Treatment-naïve and previously treated	n = 3 (TKI-naïve) n = 4 (prior TKI)	Allowed if asymptomatic; asymptomatic LMD allowed	RECIST 1.1	TKI-naïve: CNS ORR 100% Prior TKI: CNS ORR 50%	[93]
	MET ex14 alteration	GEOMETRY mono-1	capmatinib	II	Treatment-naïve and previously treated	n = 13	BrM allowed if no increase in steroid dose within 2 weeks before enrollment	RECIST 1.1	CNS ORR 54%; 3 of 7 patients with a CNS response received RT	[94]
		VISION	tepotinib	II	Treatment-naïve and previously treated	n = 15 (BrM) n = 7 (measurable BrM)	Asymptomatic BrM were allowed.	RANO-BM	Measurable BrM: CNS ORR 71%; CNS disease control rate 87%	[95]
Antibody-drug conjugate	HER2	DESTINY-Lung01	trastuzumab deruxtecan	II	Stage III-IV NSCLC HER2 overexpressing (Cohort 1) *HER2*-mutated (Cohort 2)	n = 29 (Cohort 1)	Active BrM excluded (defined as untreated and symptomatic or requiring therapy with steroids or anticonvulsants)	NA	Only systemic activity in pts with BrM was analyzed	[96,97]
		DESTINY-Lung-02	trastuzumab deruxtecan	II	Metastatic NSCLC with activating *HER2* mutation	n = 57	Active BrM excluded (defined as untreated and symptomatic or requiring therapy with steroids or anticonvulsants)	NA	Only systemic activity in pts with BrM was analyzed	[98]
	HER3	HERTHENA-Lung01	HER3-DXd	II	*EGFR* L858R or Ex19del NSCLC Previously treated with EGFR TKI and platinum chemotherapy	n = 30 (no prior RT)	Excluded if symptomatic/untreated or requiring therapy with corticosteroids or anticonvulsants	RECIST 1.1	CNS ORR 33.3% (95% CI 17.3–52.8%)	[99]
		HERTHENA-Lung02	HER3-DXd	III	EGFR L858R or Ex19del NSCLC Previously treated with 3rd generation TKI	Not reported	Active BrM allowed although excluded if active treatment with steroids; LMD excluded	RECIST 1.1	Not reported	[100]

**Table 3 cancers-16-03780-t003:** Selected clinical trials of immunotherapy agents in lung cancer.

Drug Class	Target(s)	Trial Name/Number	Arms	Phase	Patient Characteristics	No. of Pts with BrM	CNS Disease Inclusion Criteria	CNS Response	CNS Outcomes	Reference
Immune checkpoint inhibitors	PD-1	NCT02085070	Cohort 1: PD-L1 ≥1% + pembrolizumab Cohort 2: PD-L1 <1% (or unevaluable) + pembrolizumab	II	NSCLC cohort	n = 42	Untreated or progressing BrM allowed; 5–20 mm	Modified RECIST 1.1	Cohort 1: 88% CNS ORR Cohort 2: 0% CNS ORR Median follow-up 8.3 months	[125]
		CAP-BRAIN NCT04211090	Camrelizumab + pemetrexed/carboplatin	II	NSCLC with no prior systemic therapy *EGFR/ALK* mutation-negative	n = 45	BrM allowed if asymptomatic or symptoms controlled with dehydration therapy	Modified RECIST 1.1	Measureable BrM: CNS ORR 52.5% (95% CI 36.1–68.5%) All BrM: CNS ORR 46.7% (95% CI: 31.7–62.1%)	[129]
	PD-L1	Atezo-BRAIN	Atezolizumab + pemetrexed/carboplatin	II	NSCLC with no prior systemic therapy *EGFR/ALK* mutation-negative	n = 40	BrM allowed if asymptomatic or symptoms controlled with medical therapy 55% received dexamethasone	RANO-BM	12-week PFS 62.2% CNS ORR 42.7% (95% CI 28.1–57.9%)	[130]
	PD-1/ CTLA-4	CheckMate 227 NCT02477826	Nivolumab + ipilimumab vs. nivolumab + chemotherapy	III	Stage IV or recurrent NSCLC *EGFR/ALK* mutation-negative	n = 202	BrM allowed if previously treated and asymptomatic	Not reported	Intracranial PFS 16% (95% CI: 5–33) vs. 6% (95% CI: 1–22%)	[126]
Bispecific antibodies	MET/ EGFR	PAPILLON NCT04538664	Amivantimab +carboplatin/ pemetrexed	III	NSCLC *EGFR* exon 20 insertion	Not reported	BrM allowed if stable, asymptomatic, and not requiring steroids or anti-convulsants for ≥14 d	Not reported	Not reported	[102]
		MARIPOSA NCT04487080	Amivantimab	III	Stage III-IV NSCLC *EGFR* L858R or ex19del	Not reported	BrM allowed if stable and asymptomatic	Not reported	Not reported	[103]
		MARIPOSA-2 NCT04988295	chemotherapy vs. amivantamab/chemo vs. amivantamab/ lazertinib/chemo	III	Stage III-IV NSCLC *EGFR* L858R or ex19del +progression on osimertinib	n = 120 (chemo) n = 58 (ami./chemo.) n = 120 (ami./lazert./ chemo)	BrM allowed if stable and asymptomatic	Not reported	Not reported	[104]
		CHRYSALIS NCT02609776	Amivantimab	I	Stage III–IV *EGFR* or *MET* mutations or amplifications (ctDNA or tumor tissue)	n = 18	BrM allowed if previously treated and asymptomatic at time of screening	NA	No report of CNS responses; no difference in ORR in patients with BrM vs. patients without	[105]
		PALOMA-3 NCT05388669	Amivantimab IV vs. subcutaneous	III	NSCLC *EGFR* L858R or exon 19del Progressed on/after osimertinib or other 3rd generation TKI	n = 34 (i.v.) n = 34 (s.c.)	BrM excluded if symptomatic or progressive; LMD excluded	Not reported	Not reported	[106]
		CHRYSALIS-2 NCT04077463	Amivantimab + lazertinib +/− chemotherapy	I/IB	*EGFR*-mutated NSCLC with progression on 1st line therapy	n = 8	BrM allowed if previously treated and asymptomatic at time of screening	Not reported	Not reported	[107]
	HER2/ HER3	NCT02912949	Zenocutuzumab	I/II	Documented *NRG1* gene fusion Part 1: Solid tumors Part 2, Group F: NSCLC	Not reported	BrM allowed if previously treated and asymptomatic	Not reported	Not reported	[108]
	HER2/ PD-L1 & CTLA-4	NCT04521179	KN026 + KN046	II	HER2-positive (IHC 3+ or HER2 gene amplification) solid tumors	Not reported	BrM allowed if previously treated and asymptomatic	Not reported	Not reported	[109]
	PD-1/ CTLA-4	COMPASSION-01 NCT03261011	Cadonilimab	I	Metastatic solid tumors refractory to standard therapies or no standard therapy	Not reported	Not specified although patients receiving steroids within 14 d of drug administration were excluded	Not reported	Not reported	[110]
		NCT04172454	Cadonilimab	Ib/II	NSCLC that had previously failed platinum-based doublet chemotherapy Cohort A: immunotherapy-naïve Cohort B: primary resistance to immunotherapy Cohort C: acquired resistance to immunotherapy	Not reported	Not specified although patients receiving steroids within 14 d of drug administration were excluded	Not reported	Not reported	[111]
		NCT04646330	Cadonilimab + anlotinib	Ib/II	Stage IIIB-IV NSCLC Without *EGFR/ALK/ROS* mutations	Not reported	Active BrM excluded	Not reported	Not reported	[112]
		MAGELLAN NCT03819465	Multi-arm study Arms A4, B4: MEDI5752	Ib	Stage III NSCLC with progression on prior treatment Stage IV NSCLC untreated *EGFR*/*ALK* mutation-negative Known PD-L1 status	Not reported	Untreated BrM excluded	Not reported	Not reported	[113]
	PD-L1/ CTLA-4	ENREACH-L-01 NCT04474119	KN046 + carboplatin/paclitaxel vs. placebo + carboplatin/paclitaxel	III	Stage IV squamous NSCLC *EGFR*-mutation negative	NA	BrM and LMD allowed if treated or asymptomatic	NA	NA	NA
		NCT03838848	KN046	II	NSCLC, failure on platinum-based chemotherapy	n = 3	BrM excluded if untreated or active	Not reported	Not reported	[114]
		NCT04054531	KN046 + pemetrexed (non-squamous) KN046 + carboplatin (squamous)	II	Stage IV NSCLC, treatment-naïve	n = 15	BrM excluded if untreated or symptomatic	Not reported	Not reported	[115]
	PD-1/ TIM-3	NCT04931654	AZD7789	I/II	Stage IIIB-IV NSCLC (Part A) Documented PD-L1 expression EGFR/ALK mutation negative	Not reported	Symptomatic BrM or LMD excluded	NA	NA	NA
	PD-1/ TIGIT	ARTEMIDE-01 NCT04995523	Rilvegostomig	I/II	Stage III-IV NSCLC Documented PD-L1 expression *EGFR*/*ALK* mutation negative	22.5%	Symptomatic BrM excluded	Not reported	Not reported	[116]
	PD1/ VEGF	HARMONi-5 NCT04900363	Ivonescimab	1b	Stage IIIB-IV NSCLC Immunotherapy-naïve	Not reported	Active BrM excluded; note: also excluded pateints receiving steroids in the past 2 years	Not reported	Not reported	[117]
	CD3/ CEA	NCT02324257	RO6958688	I	Confirmed CEA expression in tissue (non-CRC patients) Locally advanced and/or metastatic solid tumor	Not reported	BrM allowed if previously treated, asymptomatic, and not requiring steroids or anti-convulsants for ≥14 d	NA	NA	NA
		NCT026501713	RO6958688 + atezolizumab	Ib	Confirmed CEA expression in tissue (non-CRC patients) Locally advanced and/or metastatic solid tumor	Not reported	BrM excluded if active or untreated, including if identified on CT/MRI during screening	NA	NA	NA
		Morpheus-Lung (NCT03337698)	Numerous arms, including: RO6958688 + atezolizumab	I/II	NSCLC Cohort 1: No prior systemic therapy for metastatic NSCLC, PD-L1 TPS ≥ 50% PD	Not reported	BrM excluded if symptomatic, untreated, or actively progressing; LMD excluded	NA	NA	[118]
	CD3/ PSMA	NCT04496674	Ab CC-1 + toczilizumab	I/II	Lung SCC PSMA ≥ 10% of tumor cells	NA	BrM and LMD excluded	NA	NA	NA
		NCT04822298	AMG 160	Ib	NSCLC Detectable PSMA by PET/CT	Not reported	BrM and LMD allowed if treated or asymptomatic	NA	NA	NA

## 4. Combinatorial Strategies

In EGFR-mutated NSCLC, Magnuson et al. performed a multi-institutional analysis to evaluate multidisciplinary optimal management options for patients with BrMs managed with the first-generation TKI erlotinib [142]. The patients were treated with SRS or WBRT followed by EGFR-TKI or an upfront EGFR-TKI alone, reserving SRS or WBRT for subsequent intracranial progression. The upfront SRS followed by TKI was associated with a prolonged OS (46 months) relative to the upfront WBRT (30 months) or TKI alone (26 months, *p* = 0.001). Retrospective studies in the era of newer-generation TKIs with enhanced CNS activity have exhibited varying results. One study comparing a CNS-penetrant TKI alone to a CNS-penetrant TKI with CNS radiation therapy (RT) in *EGFR*- and *ALK*-mutated NSCLC found no significant differences in the time to progression or time to intracranial progression between the TKI alone or with CNS RT [143]. In contrast, a Japanese study investigated osimertinib alone or with upfront CNS RT for BrMs from EGFR-mutated NSCLC and reported that both the OS and intracranial PFS were significantly longer with upfront RT compared to with osimertinib alone [144]. The TURBO-NSCLC retrospective study of patients treated at seven US centers examined CNS-penetrant TKIs with or without upfront SRS in patients with *EGFR* or *ALK* alterations [145]. The patients receiving upfront SRS were more likely to have at least one BrM measuring 1 cm or larger (*p* < 0.001) and neurologic symptoms (*p* < 0.001) at initial presentation. The upfront SRS was associated with improvements in the local CNS control (HR 0.30 [95% CI 0.16 to 0.55], *p* < 0.001) and time to CNS progression ([HR], 0.63 [95% CI, 0.42 to 0.96]; *p* = 0.033), whereas there were no differences in the OS. The patients with larger BrMs (≥1 cm) had higher rates of CNS progression, and the subgroup analyses suggested that these patients may benefit more from upfront SRS [145].

A strategy of an upfront TKI to reduce the burden of CNS disease and avoid WBRT, potentially followed by SRS, termed “CNS downstaging”, has also being examined as a strategy for *EGFR*-, *ALK*-, or *ROS1*-driven NSCLC patients with extensive BrMs (>10) at presentation [146]. In an initial study from the University of Colorado, 12 patients with a median of 49 BrMs at presentation were treated with upfront CNS-penetrant TKI, resulting in a median 97% reduction in the CNS tumor volume per patient and a median of five residual BrMs at the best CNS response. All the patients avoided WBRT and seven patients subsequently received SRS. This case series suggests that the upfront administration of CNS-active TKIs could represent a promising strategy to downstage extensive CNS presentations and to convert some patients into SRS candidates. Phase II trials are ongoing investigating Osimertinib with or without upfront SRS (NCT03497767, NCT03769103), which have recently been reported in abstract. At 12 months, the intracranial PFS was 70% (95% CI 52–93%) for upfront SRS and 53% (95% CI 34–81%) for upfront osimertinib in LUOSICNS (NCT03769103) and 67% (95% CI 49–93%) with upfront SRS vs. 67% (95% CI 49–90%) for upfront osimertinib in OUTRUN (NCT03497767) [147]. There was no statistically significant difference between the two arms in the intracranial PFS or overall survival. Prospective trials are evaluating outcomes following the randomization of patients with oncogene-mutated NSCLC and asymptomatic BrMs to upfront or delayed cranial RT (SRS or WBRT) (NCT05236946, ICON-RT) and will provide more insight into the timing of SRS in the context of systemic osimertinib.

Given the potential modulating effects of radiation on the anti-tumor immune response [148], there are numerous ongoing studies evaluating the safety and efficacy of combining RT and ICIs. One retrospective analysis examined the activity and toxicity profile of pembrolizumab in patients who had received previous RT (extracranial or thoracic) in the KEYNOTE-001 phase I trial [149]. In the patients who had previously received any RT, pembrolizumab led to an improved median survival compared to that in patients without prior RT (10.7 vs. 5.3 months, *p* = 0.026). Moreover, the PACIFIC trial demonstrated significantly longer survival in stage III NSCLC patients with the addition of durvalumab to chemoradiotherapy with a lower incidence of new BrMs with durvalumab (5.5% vs. 11.0%) [150]. In regard to the safety of CNS-directed therapies, a retrospective study of patients with newly diagnosed BrMs treated with SRS, including 294 NSCLC patients, found symptomatic radiation necrosis in 20% of patients who had received ICIs compared to 6.8% of patients who had not received ICIs [151]. Phase I/II trials have demonstrated that SRS given with nivolumab/ipilimumab or pembrolizumab has an acceptable safety profile and favorable outcomes [152,153,154]. However, a systematic review of 16 studies found a trend towards increased incidence of radionecrosis with SRS and ICIs compared to SRS alone (16.0% vs. 6.5%, *p* = 0.065) [155]. An IRRF study of 395 patients with 2540 BrMs found that the risk of any-grade radionecrosis and symptomatic radionecrosis following single-fraction SRS increased as V12 Gy exceeds 10 cm^3^ and was comparable between concurrent and non-concurrent SRS and ICIs [156]. These results suggest that the combination of SRS and ICIs may lead to the increased incidence of radionecrosis, and prospective studies are needed to elucidate the safety and efficacy of combining these modalities [157]. Prospective trials examining the combination of SRS and ICIs are ongoing (NCT03340129, NCT04889066, NCT05522660, NCT06501391) and will elucidate if this combinatorial strategy can yield durable outcomes in NSCLC patients with BrMs.

## 5. Leptomeningeal Metastases

Leptomeningeal metastasis (LM), or the spread of cancer to the cerebrospinal fluid (CSF) and leptomeninges, harbors a very poor prognosis. In addition to systemic therapy strategies, LM has historically been treated with radiation for symptomatic disease, WBRT, or, in highly selective cases, cerebrospinal irradiation (CSI). LM presentations are heterogeneous and treatments may be highly individualized. Both WBRT and involved field radiation of symptomatic sites in the brain or spine are commonly used in clinical practice [9]. Photon-based CSI using three-dimensional conformal radiation therapy (3D CRT) has demonstrated improved neurologic function and survival in patients with LM, but also led to significant toxicity to the esophagus, bowel, and bone marrow [158]. Given the toxicity with photon CSI, hypofractionated proton CSI (pCSI) was evaluated in patients with LM and dose-limiting toxicities were found of lymphopenia, thrombocytopenia and fatigue and durable control in 4 of 21 patients, who were free of CNS progression for more than 12 months [159]. A phase II trial subsequently randomized patients with NSCLC or breast cancer and LM 2:1 to pCSI or photon IFRT and found the primary endpoint of CNS PFS to be significantly improved with pCSI versus IFRT (median 7.3 months vs. 2.3 months, *p* < 0.001) [160]. The NRG phase III RADIATE-LM trial is ongoing to investigate outcomes with pCSI compared to IFRT in breast and NSCLC patients with LM using IMRT (NCT06500481).

Recent developments in systemic therapies have contributed to improved outcomes in LM. A retrospective review of pemetrexed in the treatment of LM in patients with *EGFR*-mutated NSCLC demonstrated prolonged survival with pemetrexed use after the development of LM relative to no pemetrexed use (median 13.7 months vs. 4.0 months, *p* = 0.008). However, prospective studies are needed to investigate if systemic chemotherapy can indeed prolong survival in NSCLC patients with LM. CNS-penetrant TKIs have demonstrated activity in LM in patients with oncogene-driven NSCLC, including alectinib in *ALK* fusion-positive [161], selpercatinib in *RET*-rearranged [162], and osimertinib in *EGFR*-mutated [163,164]. In prospective trials of osimertinib in LM in patients with *EGFR*-mutated NSCLC that had progressed with prior *EGFR*-directed TKI, the LM ORR was 41% with 160 mg daily [163] and 51.6% with 80 mg daily [164]. A retrospective multi-institutional study of patients with LM treated with osimertinib 160 mg daily found that among patients with CNS progression on 80 mg daily, dose escalation to 160 mg led to a median CNS control of 3.8 months, which was increased to 5.1 months with concurrent chemotherapy and/or radiation [165]. A prospective trial treated patients with *EGFR*-mutated NSCLC and LM with osimertinib and bevacizumab, achieving an ORR of 50% and a median OS of 12.6 months [166]. Thus, combinatorial strategies with osimertinib represent a promising strategy for the management of LM.

Intrathecal chemotherapy (ITC) has recently been evaluated in NSCLC patients with LM. A pooled analysis was performed of four prospective and five retrospective studies comprising 552 patients; 37 patients received ITC only and achieved a median survival of 7.5 months. The majority of the patients received a multimodal approach (e.g., ITC and SRS) with a median survival of 3.0 to 5.0 months; which could possibly reflect the higher toxicity of combining therapeutic approaches in patients with LM [167]. A phase II trial of intrathecal pemetrexed in patients with *EGFR*-mutated NSCLC who had progressed on TKIs demonstrated a response rate of 80.3% with acceptable side effects [168]. Further studies are needed to assess the optimal multi-intervention approach to LM.

## 6. Concluding Remarks

In conclusion, the management of brain metastasis in lung cancer is increasingly personalized. Stereotactic radiosurgery is preferred for up to 10 brain metastases and should be considered for up to 20–30 at the discretion of the treating radiation oncologist [27]. For patients with actionable mutations with CNS-active therapies, multidisciplinary decision making should guide the timing of systemic therapies with SRS or WBRT as appropriate [27]. Increasing systemic therapies are available with CNS activity that can be leveraged to reduce the radiation volume in patients with BrMs. In the case of extensive BrMs, our institutional experience has been to use several approaches depending on the systemic disease, patient factors, and patient preference, including WBRT, WBRT followed by SRS to non-responding lesions, and SRS to larger lesions when patients are on CNS-active therapy (Figure 1). Further prospective randomized trials are needed to guide multidisciplinary management in these various contexts.

## 7. Future Directions

Collectively, the management of BrMs centers around balancing individual patient factors, e.g., performance status and preference, with preserving neurocognition both by curbing tumor progression as well as sparing healthy brain tissue. The emergence of CNS-penetrant targeted therapies shows promise for providing local control. In addition, the FLAURA2 [120] and CAP-BRAIN [129] trials suggest that combinatorial strategies could overcome drug resistance mechanisms to yield durable intracranial responses and prevent morbid CNS failures. However, the durability of local control from targeted therapies is unclear and the anti-tumor efficacy must be balanced against adverse side effects. The optimal integration of radiotherapy with systemic approaches thus represents an evolving question in the management of BrMs in lung cancer. Studies evaluating the biologic mechanisms of radiation and systemic therapies are needed to elucidate possible synergistic effects, for example, between the immunogenicity of radiation and immune checkpoint inhibitors [156].

It will be crucial to determine through prospective randomized trials which patients may benefit from the combination of SRS and systemic agents vs. de-intensification strategies. Ongoing research regarding predictors of synergistic toxicities between emerging systemic therapies and CNS-directed RT are essential to individualizing patient management. For patients with BrMs from NSCLC with targetable mutations, the ICON-RT trial will investigate a strategy of using upfront CNS-active drug therapy to debulk (or downstage) intracranial disease followed by individualized treatment only to residual or progressive lesions with SRS. This trial will evaluate whether patients with high-risk lesions harboring drug resistance mechanisms might benefit from therapeutic intensification with radiation.

Additionally, advances in radiotherapy, including increasingly targeted radiation techniques, e.g., IMRT over 3D CRT, in combination with approaches to preserve cognitive function, e.g., memantine or hippocampal avoidance, can be leveraged to balance the local control afforded by radiotherapy with the possibility of cognitive decline or radionecrosis. Clinical trials are needed to determine if strategies such as TTFs or the integration of RT with systemic therapies may avoid WBRT in patients with extensive BrMs at baseline.

Another integral aspect of the management of lung cancer BrMs includes predicting which patients will develop BrMs to guide clinical decision making. Radiomics have recently been described as a strategy to predict BrMs in T1 or curatively resected lung adenocarcinoma [170,171]. Cerebrospinal fluid (CSF) circulating tumor cells (CTCs) have been found to be useful both in diagnosing LM [172] as well as in predicting response to pCSI [173]. Cell-free DNA (cfDNA) has been detected in the CSF and has been proposed as a liquid biopsy for monitoring CNS metastases [174]. In addition, cfDNA signatures were found to have a high negative predictive value (85%) and specificity (93%) in excluding BrM development in breast cancer. In lung cancer, unique profiles of circulating tumor DNA (ctDNA) were associated with the development of BrMs and changes in the ctDNA within the CSF could predict intracranial responses [175]. Prospective studies are needed to inform if CSF or peripheral biomarkers can guide the treatment of BrMs. Indeed, prospective studies are underway to sample CSF for CTCs and cfDNA (NCT04343573).

Another approach is evaluating the genomic or histological features of the primary tumor. *CDKN2A/B* deletions and alterations in the cell cycle pathway have been found to be associated with BrMs while *MYC* amplifications have been found to be associated with multifocal BrMs [176]. Given that *MYC* alterations are common in SCLC [6], it is possible that *MYC* mutations are associated with the pattern of BrMs in SCLC and this warrants further investigation. In addition, the expression of ASCL1 or NEUROD1 on immunohistochemistry, as biomarkers associated with BrMs, was associated with worse CNS outcomes in BrM-free SCLC patients at diagnosis [177]. As biomarkers associated with BrMs are increasingly identified, prospective studies are needed to guide personalized approaches leveraging these findings.

Efforts to predict response have attempted to integrate various features such as imaging, clinical features, and genomic information. A CoxCC model for progression risk stratification of EGFR TKI treatment integrated MR radiomics, clinical features, and a “prognostic index” using molecular markers (lung-molGPA), and the modified lung-molGPA achieved an AUC of 0.88, 0.73, 0.92, and 0.90 for predicting the PFS at 3, 6, 9, and 12 months, respectively [178]. Given the advances in supervised and unsupervised machine learning, further studies to develop predictive models for BrMs in both NSCLC and SCLC patients are needed to guide therapeutic approaches.

Collectively, as advances are made in CNS-penetrant systemic therapies, the integration of radiation with multimodality regimens, and the identification of predictive biomarkers, the management of BrMs will become increasingly personalized. It will be critical to leverage level I evidence to define how best to optimize management. The selection of a multidisciplinary treatment should take into account the CNS disease, overall clinical picture, and patient preference regarding therapeutic tradeoffs, such as intracranial disease control and quality of life.

## Figures and Tables

**Figure 1 cancers-16-03780-f001:**
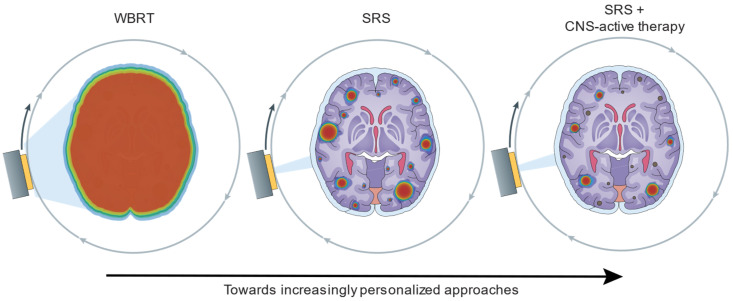
Towards personalized therapeutic strategies in lung cancer patients with BrMs. Due to the hematogenous nature of the dissemination of BrMs, patients historically received WBRT for the treatment of multiple BrMs (**left**). However, due to clinical trials demonstrating neurocognitive toxicity with WBRT and efficacy of SRS for numerous BrMs, SRS has become the standard of care for up to 10 BrMs and ongoing trials are evaluating the efficacy of SRS up to 30 BrMs (**center**). In the era of novel targeted therapies and combinatorial strategies with intracranial efficacy, the treatment of BrMs is becoming increasingly personalized (**right**), with the use of SRS to target persistent metastases or those with poor response to systemic therapies. Images adapted from Figure 3a of Suh et al. [169].

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
