# Peer review of "Advances in the Management of Lung Cancer Brain Metastases"

_cancers, 2024, doi:10.3390/cancers16223780_

Round 1
Reviewer 1 Report
Comments and Suggestions for Authors
In this very comprehensive review the authors summarize the evidence on brain metastases (BM) management in a multidisciplinary context. This includes radiotherapy, the mainstay in the treatment of BM, as well as systemic therapies with a focus on immunotherapy and targeted drugs in case of driver mutations, combined approaches and finally leptomeningeal metastatic disease. This review includes a huge amount of information, which is presented in a clearly structured manner. The manuscript is well written so that the lines of thought are easy to follow.
I believe the following aspects should be considered before publication as they could possibly enhance the quality of the paper.
1. The studies in table 1 – if they refer to NSCLC – contain a considerable number of non-small cell lung cancer patients. These studies, e.g. Aoyama (reference 20), Chang (reference 21), Kocher (reference 22), Brown (reference 23) and other contain 50% to 70% BM from lung cancer. Other entities included are breast, prostate, kidney and colorectal cancer. Only 2 studies (QUARTZ and METIS) were conducted in NSCLC exclusively. This mixture of primary tumor sites may have an influence on outcome in as far as – for example – BM several years after breast cancer diagnosis is clinically different from BM in NSCLC, which often occurs simultaneously with the diagnosis of primary disease in the lung.
· I think that this multiplicity of entities in the studies should be discussed in the manuscript.
· With reference to the title, which is specifically BM from lung cancer: Is it possible to separate the results on BM due to NSCLC from the rest of the patients?
2. As said above, this manuscript presents a lot of information on the current state-of-the-art treatment of BM in lung cancer (and other entities). To my mind, concluding recommendations at the end are missing so that the authors may want to include a paragraph before Future directions on how BM should be treated.
Author Response
In this very comprehensive review the authors summarize the evidence on brain metastases (BM) management in a multidisciplinary context. This includes radiotherapy, the mainstay in the treatment of BM, as well as systemic therapies with a focus on immunotherapy and targeted drugs in case of driver mutations, combined approaches and finally leptomeningeal metastatic disease. This review includes a huge amount of information, which is presented in a clearly structured manner. The manuscript is well written so that the lines of thought are easy to follow.
We thank the reviewer for their summary and favorable receipt of our manuscript.
I believe the following aspects should be considered before publication as they could possibly enhance the quality of the paper.
- The studies in table 1 – if they refer to NSCLC – contain a considerable number of non-small cell lung cancer patients. These studies, e.g. Aoyama (reference 20), Chang (reference 21), Kocher (reference 22), Brown (reference 23) and other contain 50% to 70% BM from lung cancer. Other entities included are breast, prostate, kidney and colorectal cancer. Only 2 studies (QUARTZ and METIS) were conducted in NSCLC exclusively. This mixture of primary tumor sites may have an influence on outcome in as far as – for example – BM several years after breast cancer diagnosis is clinically different from BM in NSCLC, which often occurs simultaneously with the diagnosis of primary disease in the lung.
- I think that this multiplicity of entities in the studies should be discussed in the manuscript.
We thank the reviewer for raising the above points. We have included wording in the text to clarify that these trials include other histologies, and we include the proportion of lung cancer in the clinical trials in Table 1.
- With reference to the title, which is specifically BM from lung cancer: Is it possible to separate the results on BM due to NSCLC from the rest of the patients?
While a number of the radiation therapy clinical trials selected in this manuscript for the NSCLC encompass other histologies, decisions around SRS and WBRT are typically made treatment recommendations for radiation therapy. We thus believe that delineating NSCLC outcomes only would not significantly change the meaning and conclusions of the radiation section in contrast to the systemic therapy clinical trials, which are exclusively in lung cancer patients.
- As said above, this manuscript presents a lot of information on the current state-of-the-art treatment of BM in lung cancer (and other entities). To my mind, concluding recommendations at the end are missing so that the authors may want to include a paragraph before Future directions on how BM should be treated.
We thank the reviewer for this suggestion. We have included a paragraph on concluding remarks prior to the Future Directions section. In addition, we cite the NCCN guidelines for further exploration for readers.
Reviewer 2 Report
Comments and Suggestions for Authors
Well-written review paper on advances in the management of patients with brain metastases from non-small cell and small cell lung cancers. Suggest the following additional information:
Lines 78-84: please include references and/or the names of the ongoing clinical trials mentioned in this section.
Line 177-178: 21 Gy in 1 fraction was not one of the dose levels in RTOG 9005. Can you clarify regarding that specific dose ( as opposed to 24 Gy for 2 cm or less)
Line 256-257: add two more pre-op vs. post-op studies NCT 05438212 and NCT05871307
Section 2.3: would add mention that there is a limited role of surgery in SCLC brain metastases.
Line 558-560: add reference on the pemetrexed LMD review.
Under systemic therapy section for SCLC (or could add to discussion in combination strategies section) mention potential intracranial activity of tarlatamab but acknowledging that the patients enrolled had treated brain metastases (PMID: 39208379).
Author Response
Well-written review paper on advances in the management of patients with brain metastases from non-small cell and small cell lung cancers. Suggest the following additional information:
We thank the reviewer for their favorable receipt of our manuscript.
Lines 78-84: please include references and/or the names of the ongoing clinical trials mentioned in this section.
We have included citations for all clinical trials that are mentioned in these lines. Please clarify if there is something specific that should be added.
Line 177-178: 21 Gy in 1 fraction was not one of the dose levels in RTOG 9005. Can you clarify regarding that specific dose ( as opposed to 24 Gy for 2 cm or less)
RTOG 90-05 used 18 Gy in 1 fraction, 21 Gy in 1 fraction, and 24 Gy in 1 fraction for tumors <2 cm. Our institutional preference is for 21Gy in 1 fraction based on adequate local control rates with acceptable radionecrosis rates. We have updated the language to say that 15-24Gy are used in clinical practice depending on patient- and tumor-specific factors.
Line 256-257: add two more pre-op vs. post-op studies NCT05438212 and NCT05871307
We thank the reviewer for pointing out these clinical trials and have added them to the manuscript.
Section 2.3: would add mention that there is a limited role of surgery in SCLC brain metastases.
We thank the reviewer for this point. The indication for surgery in NSCLC or SCLC is typically for the management of large and/or symptomatic brain metastases for the management of mass effect or symptoms or for diagnostic information. We have revised the wording to clarify that that aspect is more general. We have added that there is limited data on the efficacy of surgery with radiation for the management of brain metastases.
Line 558-560: add reference on the pemetrexed LMD review.
We thank the reviewer for noting this oversight and have revised the text to include the citation.
Under systemic therapy section for SCLC (or could add to discussion in combination strategies section) mention potential intracranial activity of tarlatamab but acknowledging that the patients enrolled had treated brain metastases (PMID: 39208379).
We thank the reviewer for raising this point and have added tarlatamab to the SCLC systemic therapies section.
Reviewer 3 Report
Comments and Suggestions for Authors
This review is interesting, but there are some points to revise before publishing.
1. There is no title of tables.
2. With the advancement of drug therapy, the number of cases of long-term survival is increasing, and the demand for re-irradiation (including re-SRS, re-SRT, and re-WBRT) is also on the rise. I think that if you could provide a separate section on this and provide more details, it would be more useful for readers.
Author Response
This review is interesting, but there are some points to revise before publishing.
- There is no title of tables.
We thank the reviewer for raising this point. We have added table titles.
- With the advancement of drug therapy, the number of cases of long-term survival is increasing, and the demand for re-irradiation (including re-SRS, re-SRT, and re-WBRT) is also on the rise. I think that if you could provide a separate section on this and provide more details, it would be more useful for readers.
We thank the reviewer for this suggestion, and we have included a section on re-irradiation.
Reviewer 4 Report
Comments and Suggestions for Authors
This is a good paper for readers to understand the Brain metastasis caused by lung cancer .Everything about lung cancer brain mestatasis is in detail and in order . I only hope the author could add some references about WBI+simultaneous integrated boost for tumor .
Author Response
This is a good paper for readers to understand the Brain metastasis caused by lung cancer .Everything about lung cancer brain mestatasis is in detail and in order . I only hope the author could add some references about WBI+simultaneous integrated boost for tumor.
The original clinical trials evaluating SRS compared it against WBRT + SRS (Aoyama et al, Change et al, Brown et al, Kocher et al). Similarly, RTOG 9508 compared WBRT + SRS to WBRT alone. Collectively, these trials as well as a meta-analysis (Tsao et al, 2018) have demonstrated that WBRT + SRS provides improved intracranial control, but no benefit in overall survival and the patients receiving WBRT experienced neurocognitive decline. We include this discussion in a concluding paragraph on recommendations. We also discuss our institutional experience with WBRT followed by SRS to non-responding lesions and observing small lesions on CNS-active therapy while providing SRS to lesions >1cm. The ICON-RT phase II randomized trial is assessing the latter with osimertinib.
Round 2
Reviewer 1 Report
Comments and Suggestions for Authors
I have no further comments.